# Literature Review on Power Battery Echelon Reuse and Recycling from a Circular Economy Perspective

**DOI:** 10.3390/ijerph20054346

**Published:** 2023-02-28

**Authors:** Yongyou Nie, Yuhan Wang, Lu Li, Haolan Liao

**Affiliations:** 1School of Economics, Shanghai University, 99 Shangda Road, Baoshan District, Shanghai 200444, China; 2College of Environmental Science Engineering, Hunan University, Changsha 410082, China

**Keywords:** EoL power batteries, echelon utilization, circular economy, recycle methods, recycle system

## Abstract

Developing new energy vehicles (NEVs) is necessary to grow the low-carbon vehicle industry. Many concentrated end-of-life (EoL) power batteries will cause large-scale environmental pollution and safety accidents when the time comes to replace the first generation of batteries if improper recycling and disposal methods are utilized. Significant negative externalities will result for the environment and other economic entities. When recycling EoL power batteries, some countries need to solve problems about lower recycling rates, unclear division of echelon utilization scenarios, and incomplete recycling systems. Therefore, this paper first analyzes representative countries’ power battery recycling policies and finds out the reasons for the low recycling rate in some countries. It is also found that echelon utilization is the critical link to EoL power battery recycling. Secondly, this paper summarizes the existing recycling models and systems to form a complete closed-loop recycling process from the two stages of consumer recycling and corporate disposal of batteries. The policies and recycling technologies are highly concerned with echelon utilization, but few studies focus on analyzing application scenarios of echelon utilization. Therefore, this paper combines cases to delineate the echelon utilization scenarios clearly. Based on this, the 4R EoL power battery recycling system is proposed, which improves the existing recycling system and can recycle EoL power batteries efficiently. Finally, this paper analyzes the existing policy problems and existing technical challenges. Based on the actual situation and future development trends, we propose development suggestions from the government, enterprises, and consumers to achieve the maximum reused of EoL power batteries.

## 1. Introduction

Under the dual pressure of global ecological degradation and the energy crisis, the world landscape is undergoing significant and profound changes. The global transportation sector will consume 60% of total oil consumption, contributing 26% of carbon dioxide emissions, and has become the major source of urban air pollution [1,2,3,4]. As long as large amounts of greenhouse gases, primarily carbon dioxide, are emitted, we will meet environmental problems such as global warming, which will directly threaten the ecological environment on which we depend [5,6]. As a result of conventional fuel vehicles using non-renewable resources as fuel, greenhouse gas emissions will increase and environmental damage will be irreversible. For example, Chinese crude oil import dependency (73%) and carbon emissions (9899 million tonnes) reached record highs in the same year (see Figure 1). Therefore, a growing number of countries are proposing net zero emission (NZE) targets and integrating them with sustainable development, energy security, and urban sustainability beyond the limits of their global carbon budgets [7,8,9]. From the perspective of the future development direction of the transport industry, the model of green sustainability, energy conservation, and emission reduction has become the core support point of various national strategies. Sustainable mobility will become the focus of future transportation industry [10]. The electric car market is one of the most dynamic in clean energy. Climate change, air pollution, and oil import dependency can all be addressed by electrifying transportation [11]. The promotion of NEVs can reduce the dependence of vehicles on fossil fuels and also mitigate major environmental issues such as “carbon emissions” and “greenhouse effect” [12,13]. Vigorously promoting the development of NEVs is the only way to promote sustainable economic growth.

A strategic emerging industry, the NEV industry has received substantial support from government [14]. In 2021, NEV sales doubled from the previous year to 6.6 million. The number of NEVs used for transportation worldwide is predicted to surpass 300 million by 2030 [15]. China, the EU, and the U.S. accounted for nearly two-thirds of global electric car sales. In recent years, China has played a significant role in the development of the global NEV industry [16]. China’s 14th five-year plan (2021–2025) emphasizes “strategic emerging industries”, including the NEV industry. As a result of intensive support policies and high financial subsidies, the NEV industry has also been placed in a highly strategic position at the national level. Figure 1 illustrates the sales volume and growth rate of NEVs in China over the past ten years. It is expected that by 2030, the proportion of new energy and clean energy-powered transport vehicles will be around 40% (refer to “the action plan for carbon peak before 2030” enacted by The State Council of China). It is undeniable that NEVs have had a revolutionary impact on the traditional automotive industry, and the electrification of automotive products has shown irreversible development [17]. With the popularity of NEVs, the amount of power battery scrap is expected to grow exponentially. As an example, Figure 2 shows the predicted amount of scrap power batteries in China [18]. Based on the average lifespan of power batteries of 5 to 8 years, the first generation of NEVs is also approaching the time when they need to replace their power batteries. EoL power batteries are convenient but have many problems. On the one hand, as the power battery market continues to expand, it is important to ensure a stable supply of raw materials. Lithium, cobalt, and nickel are three of the most critical metals for batteries. New mines are not being built fast enough to keep up with the extraordinary rise in demand for batteries. Rising prices for these metals have led to fears of a shortage in many countries [19,20]. On the other hand, the existing NEV industry mostly adopts the open-loop production model of “cradle-to-grave”, where the industrial ecological chain ends at the EoL node. In the absence of proper disposal of EoL power batteries, they will adversely affect the ecological environment, economic development, and social governance, which is contrary to the strategic goal of developing NEVs and restricts the sustainable development of the NEV industry to a great extent [21,22]. Improper recycling and disposal methods could result in secondary pollution and safety incidents, with substantial negative externalities on the ecological environment and other economic entities.

In order to reduce the negative externalities of EoL power batteries to the environment and relieve the pressure caused by insufficient raw materials, there has been widespread attention given to the issue of reuse and recycling of EoL power batteries. Existing studies have also confirmed the necessity and feasibility of the economic, environmental, and resource benefits [23]. Neubauer et al. [24] found that second-use batteries in energy storage devices can extend their lifetime, reducing the cost of producing NEVs and storing energy. Hao et al. [25] found that effective recycling of EoL power batteries could reduce greenhouse gas emissions by 6.62% and energy consumption by 8.55% to achieve additional environmental benefits. Feng et al. [26] proved that recycling EoL power batteries could reduce environmental and resource problems by 5–30% and is a crucial way to achieve sustainable development. Wang et al. [27] found that direct recycling of used lithium iron phosphate batteries can save more than 30% of metal resources and effectively alleviate the problem of resource shortage in countries around the world. At present, research on power battery recycling is focused on echelon utilization and resource recovery [28]. However, despite the macro policy and specific implementation of enterprise-level power battery recycling, there are still some problems that need to be improved.

(1)The recycling rate in some countries is relatively low.(2)Countries worldwide are exploring the echelon utilization of EoL power batteries. However, the specific application scenarios and implementation process need to be clearly delineated.(3)To guide remanufacturing enterprises toward a sustainable direction, policies for recycling EoL power batteries should be formulated from a systems perspective in some countries.

Therefore, it is highly significant to sort through current policies, recycling methods, and systems to classify different application scenarios and propose reasonable recycling systems for EoL power batteries. These actions will help to address the challenges in EoL power battery recycling operations and improve laws and regulations related to the recycling process.

This paper is different from the above-mentioned literature studies as it provides a detailed overview of the power battery recycling policies of typical countries and analyzes the characteristics of countries with high recycling rates. Second, this paper provides a comprehensive summary of the existing recycling models and systems for power batteries to fill the gap in this field. Third, this paper clearly divides the application fields of power battery gradient utilization to clarify the specific classification. Fourth, this paper proposes the 4R power battery recycling system in the context of circular economy to provide a theoretical basis for industrial development.

## 2. National Regulations for Recycling EoL Power Batteries

A country’s policies and regulations play a crucial role in developing emerging industries. Some countries have begun to pay attention to recycling EoL power batteries and have introduced many regulations. By analyzing the policies of the EU, U.S., Japan, and China, we summarize the policy characteristics of different regions and propose the development trend of recycling EoL power batteries.

### 2.1. EU

In 1991, the EU adopted its first Battery Directive (Council Directive 91/157/EEC), which emphasized the reduction of toxic emissions [29]. The End-of-Life Vehicles Directive 2000/53/EC was enacted in 2000 to encourage manufacturers and importers to manage EoL vehicles by stricter environmental standards [30]. In 2006, Battery Directive 2006/66/EC explicitly prohibited all forms of battery disposal and incineration [31]. This directive requires that batteries for EVs be collected separately and that their storage and disposal be monitored [30]. Battery Directive 2006/66/EC emphasizes the extended producer responsibility (EPR) for battery producers as well as the importance of recycling batteries and resource recovery [29]. Moreover, this directive requires at least 45% of the EoL lithium-ion batteries to be collected and at least 50% of the collected batteries to be recycled [32]. The Battery Directive is the EU’s current battery-specific legislation, primarily covering three types of batteries: portable, automotive, and industrial batteries. Among them, power batteries are classified as industrial batteries [22]. In 2013, the EU issued the New Battery Directive 2013/56/EU, which removes the restriction on cadmium in radio tool batteries and prohibits mercury in button batteries. With the adoption of Battery Directive 2006/66/EC in 2017, the European Commission, EU member states, industry, and the scientific community established the European Battery Alliance [33]. However, according to current developments in the EU, Battery Directive 2006/66/EC has become obsolete and there is an urgent need to revise the new directive in line with technological and industrial developments [22]. These battery directives do not address the environmental risks that batteries pose throughout their life cycles. The existing battery directive was developed primarily for portable battery recycling. Although it increases the overall recycling efficiency of lithium-ion batteries, its effect on the recycling efficiency of power batteries is minimal. In 2020, the EU published the new EU battery regulation 2020/0353 (COD) and proposed repealing Battery Directive 2006/66/EC, which was implemented gradually starting on January 1, 2022. The new Battery Regulation requires companies to track the life cycles of batteries placed on the market to ensure safe use and high performance [22]. As the market for electric vehicles (EVs) continues to grow, the EU has proposed a new classification for EV batteries. The regulation retains the current battery directive’s restrictions on cadmium and mercury, while also considering negative externalities and environmental risks at various stages of the battery’s life cycle. The regulation emphasizes carbon footprint and recycling efficiency requirements for EV batteries [22]. This regulation mandates that battery manufacturers assume primary responsibility for collecting, remanufacturing, reusing, and recycling EoL batteries. The EU has published a strategic research agenda for batteries and the Battery Innovation Roadmap 2030, which mandates that all battery manufacturers register and assume responsibility for the recycling and echelon utilization of batteries [34]. Table 1 lists the relevant regulations on EoL power battery recycling in EU.

EU member states have also developed regulations that are appropriate for the development of their respective industries. Germany has extended its legal framework based on EU directives. The legal system specifies the registration requirements for battery manufacturers and retailers and the recycling responsibilities of battery manufacturers, retailers, and consumers. Importers and manufacturers of batteries must register with the government before production, retailers must work with manufacturers to educate consumers on how to recycle batteries, and consumers must return used batteries to designated recycling institutions [35]. Germany has also established a foundation and deposit system and constructed many battery recycling sites for sorting and collecting batteries. These initiatives have increased consumer motivation to recycle batteries and ensured the safety of the recycling process. Volkswagen and BMW are actively implementing recycling networks for batteries. Volkswagen began participating in the Supply Chain Program in 2015 and launched a pilot project on recycling power batteries in Salzgitter [36]. Additionally, German companies have begun investigating a phased utilization of EoL power batteries. Volkswagen recycles batteries for use in energy storage and charging systems. BMW incorporates used energy storage batteries from the MINIE prototype and the BMW i3 into a scalable energy storage grid. In the Netherlands, Auto Recycling Nederland (ARN) was established due to the absence of local automotive plants. By paying a levy to ARN, automakers can allow the organization to fulfill their EPR obligations instead of handling them directly. Violating the directive can result in a hefty fine for the automaker. Therefore, the ARN can recycle the majority of (EoL) vehicles in the Netherlands in a regulated manner [37].

### 2.2. US

The U.S. was the first nation to regulate the recycling of EoL batteries with laws. As a result, the U.S. has a practical legal framework, technical specifications, and a recycling system for managing the recycling of EOL power batteries. The U.S. primarily constructs its system at the federal, state, and local levels [38]. There are distinctions between the regulation introduced at different levels of government. The federal-level emphasizes the development of industry-specific regulations and macro standards. The state-level clarifies the obligations and responsibilities of various parties in the recycling process and mandates that battery associations regulate multiple parties. Through tax credits and environmental deposit systems, the local government encourages and directs the public to participate in the recycling process, thereby promoting the implementation of the regulation. The three levels of law complement and regulate one another, making the U.S. legal system for recycling EoL batteries extremely comprehensive, detailed, and specific.

The “Resource Conservation and Recovery Act” and “Mercury-Containing and Rechargeable Battery Management Act” regulate federal recycling laws for EoL batteries. The acts mandate that battery manufacturers design batteries by keeping disassembly and recycling in mind and controlling EoL batteries in transportation, manufacturing, and recycling [31]. The state governments enact pertinent legislation based on local policies. For instance, the New York State Department of Environmental Conservation passed the“New York State Rechargeable Battery Recycling Act” in May 2010. The statute prohibits the intentional disposal of rechargeable batteries as solid waste to reduce the release of toxic metals into the environment. Since excessive production can harm the ecosystem, air quality, and human health, the policy shifted its emphasis from reducing battery toxins to resource recovery. In May 2014, Vermont enacted legislation establishing a regulatory program for primary battery products. The law gradually incorporated the EPR and clarified each entity’s recycling obligations [29]. In addition, California, New York, and Minnesota have prohibited the disposal of lithium-ion batteries in landfills [39].

The U.S. Department of Energy recognizes the critical role of EoL power batteries in the clean energy industry. It is gradually focusing on applying EoL power batteries in the field of echelon utilization in light of the increasing popularity of NEVs. The “National Lithium Development Blueprint 2021–2030”, released by the Department of Energy in 2021, proposes to maximize the use of EoL lithium batteries, establish a special fund for battery recycling, recover vital raw materials, and develop a competitive industry chain for lithium battery recycling [40]. The Department of Energy released two funding announcements in 2022: the “Battery Materials Processing and Battery Manufacturing Funding Opportunity Announcement” and the “Electric Drive Vehicle Battery Recycling and Second-Life Applications”. The U.S. provides adequate funding for constructing battery material plants, battery pack manufacturers, recycling facilities, and laddering areas. Table 2 lists the federal regulations on EoL power battery recycling in the U.S.

General Motors (GM) has recognized the potential advantages of recycling power batteries and has invested in ‘Lithion Recycling’, which has a recycling rate of over 95%. GM utilizes advanced recycling technology to recycle battery materials and co-design recyclable power cells. GM ensures a stable vehicle production supply chain and reduces production costs by investing [34].

### 2.3. Japan

The circular economy (CE) law in Japan is one of the most inclusive in the world. The Japanese legal system consists of three levels, from highest to lowest: basic law, comprehensive law, and special law, which form a product, consumption, reuse, and based circular system. The complete law framework of the CE addresses the negative externalities of batteries and provides specific operational guidance and a legal basis for battery recycling [41]. Japan began a battery recycling program in 1994 and established a system for producing, marketing, and recycling batteries due to the lack of material resources, such as coal, iron ore, and oil. In 2000, the Environment Agency issued the “Basic Law for Establishing the Recycling-Based Society”. The law regulates the basic waste and recycling policy principles, effectively promoting the 3R (reduce, reuse, and recycle) concept [42]. In the same year, the Ministry of the Environment renamed the “Law for Promotion of Utilization of Recyclable Resources”, enacted in 1991, as the “Law for Promotion of Effective Resource Utilization” following significant amendments [43]. This comprehensive law emphasizes waste reuse, including the reduction of waste generation, the reuse of EoL parts, and raw materials [44]. Due to the relatively low prices of non-ferrous metals between 1990 and 2005, the recycling of EoL vehicles would incur losses, leading some businesses to dispose of toxic waste illegally. In 2002, the government enacted an EPR-based “Law for the Recycling of End-of-life Vehicles”, which was implemented in 2005 and encouraged more companies to enter the technology research field to prevent this phenomenon. This special law establishes a new recycling system to guide companies in the proper disposal of EoL vehicles and requires consumers to pay a recycling fee when they purchase a new vehicle. Before this, Japan had a large population, limited land, and insufficient landfill space, which increased the illegal disposal of EoL vehicles by companies. Therefore, the regulation emphasized the recycling management of toxic components and residues in EoL vehicles [45]. In 2016, the Ministry of Economy, Trade, and Industry issued a “Suggestion for using Specified Recycling Deposit”, which proposed that vehicles using recycled plastics could reduce EoL vehicle recycling disposal fees. Table 3 lists the laws on EoL power battery recycling in Japan.

Japan has pioneered the commercialization of hybrid vehicles to improve fuel efficiency and reduce greenhouse gas emissions. NiMH batteries are commonly used to power hybrid vehicles, so the initial recycling policy for power batteries centered on NiMH batteries [46]. Since NiMH batteries contain numerous rare metals of high value, such as nickel and cobalt, the Japanese government and vehicle manufacturers have established a recycling system for EoL NiMH batteries. With the advancements in technology, the power batteries produced in Japan have transitioned from NiMH batteries to lithium batteries. Japanese infrastructure and systems for recycling lithium batteries are undergoing gradual improvements, and factories are being encouraged to recycle [31].

As Japan is prone to natural disasters, emergency equipment is in high demand. Nissan, Toyota, and Mitsubishi have implemented power batteries in the step-up and reuse industries. Nissan Motor and Sumitomo Corporation have established the 4R Energy joint venture. According to the remaining capacity of EoL power batteries, 4R Energy has divided the echelon utilization scenario and applied it primarily to energy storage systems and grid energy storage. Using EoL power batteries in home emergency power and energy storage devices is a viable solution pushed by battery manufacturers and vehicle manufacturers.

### 2.4. China

China needs to establish a comprehensive battery recycling model to alleviate the shortage of critical resources, such as lithium, nickel, and cobalt [47]. The policies introduced in China are mainly in principle and there are no mandatory regulatory documents. For the support policy of the power battery industry, China mainly adopts government subsidies. Table 4 lists the relevant regulations on EoL power battery recycling in China.

China has prioritized the recycling of EoL batteries since 2003. Recycling focuses on EoL rechargeable batteries and disposable button batteries. With the advancements in science and technology and the rise in popularity of electronic devices, the following policies emphasized lead–acid recycling batteries. In 2006, the National Development and Reform Commission (NDRC) proposed the first “Technical policy on recycling and utilization of automobile products” based on the system of EPR, offering the recycling of automotive product parts and components. Since 2014, the government has begun to increase its support policies for the NEV industry, resulting in its rapid development. As a result, the Chinese demand for NEV power batteries has proliferated, and the country will soon deal with an abundance of EoL batteries. However, the government should have paid more attention to recycling EoL batteries then, so no separate document existed. The policy document for the creation of NEVs merely calls for the recycling of EoL batteries.

In 2016, NDRC issued the “Technical Policy on Electric Vehicle Power Battery Recycling (2015 Vision)”. The policy implemented the EPR, directed enterprises to recycle and reuse EoL power batteries, and established upstream and downstream enterprises to form a closed-loop recycling system. Since then, China has gradually unveiled its recycling policy for power batteries. Chinese approach focuses more on recycling lithium batteries and encourages manufacturers to establish product recycling channel facilities, as lithium batteries are the most common type of battery used in power batteries [48,49]. In 2018, the Ministry of Industry and Information Technology (MIIT) promulgated the “Interim Measures for the Management of New Energy Vehicle Power Battery Recycling” and the “Interim Regulations on Traceability Management of New Energy Vehicle Power Battery Recycling”. The policy points out that battery manufacturers and comprehensive utilization enterprises should strengthen cooperation and implement the recycling principles of echelon utilization, reusing, and recycling. At the same time, the government encourages the establishment of a recycling service network for EoL power batteries and requires power battery traceability. The document emphasizes specific requirements for power battery recycling, disassembly, packaging, and transportation processes. In January 2019, the MIIT promulgated the “New Energy Vehicle Power Battery Recycling Service Network Construction and Operation Guidelines” to regulate the construction of traditional recycling service stations in China. China carried out pilot work and developed recycling implementation plans for recycling EoL power batteries in 20 provinces and cities, including Beijing, Tianjin, and Hebei Province [50]. According to the data from the MIIT, China has 47 enterprises on the white list, which have reached the standard of recycling power battery echelon utilization. Most power battery recycling enterprises are located in Guangdong Province, Hunan Province, Jiangxi Province and Zhejiang Province, as shown in Figure 3. In January 2020, MIIT revised the “New Energy Vehicle EoL Power Battery Comprehensive Utilization Industry Specification Conditions” and “New Energy Vehicle EoL Power Battery Comprehensive Utilization Industry Specification Announcement Management Interim Measures”. The previous version of the document emphasized the need for the government to regulate the development of the industry and required enterprises to strengthen the management of the EoL power batteries comprehensive utilization industry. The policy can make the EoL power battery resourceful, large-scale, high-value utilization, and improve the total utilization of resources. The latest document focuses more on the multi-purpose and multi-level reuse of used power batteries. The document specifies that comprehensive utilization includes two areas of echelon utilization and recycling. In China, a report entitled “Management Measures for the Echelon Utilization of New Energy Vehicle Power Batteries” was released in August 2021. It follows the principle of “echelon utilization before recycling” and encourages rational utilization of retired power batteries at multiple levels [40].

Based on the national industrial development policies, local governments should consider factors such as urban construction planning, regional economic development, talent pools, and echelon utilization technology to formulate targeted local policies. The policies and local governments should complement each other to form “one local, one policy” to better implement the echelon utilization of EoL power batteries [51], such as with the “Beijing, Tianjin and Hebei Region New Energy Vehicle Power Battery Recycling Pilot Implementation Plan” and the “Sichuan Province New Energy Vehicle Power Battery Battery Recycling Pilot Work Plan”. Local governments mainly promote the recycling of EoL power batteries utilizing government subsidies. For example, the Development and Reform Commission (DRC) of Shenzhen issued the “New Energy Vehicle Promotion and Application Financial Support Policy”. The policy indicates that NEV manufacturers are responsible for recycling power batteries. The policy points out that NEV manufacturers are responsible for power battery recycling and the government subsidizes recycling and processing funds according to the standard of 20 RMB/kWh. The DRC of Guangxi issued the “New Energy Vehicle Promotion and Application Three-Year Action Financial Subsidy Implementation Rules”. The government subsidizes 20 RMB/kWh according to the power battery recycling volume and the construction of recycling service stations at 30% of the cost. Hefei city government gives manufacturers a 10 RMB/kWh subsidy to support enterprises in establishing power battery recycling systems [52].

### 2.5. Policy Comparison and Analysis

By analyzing the regulations of different countries, we can observe that each country has distinct characteristics. However, all of them recognize the significant role of EPR in recycling power batteries [37]. We analyzed two key issues.

(1)Some countries face low recycling rates, with China having a lower recycling rate for end-of-life (EoL) power batteries compared to other countries. High recycling rates are typically observed in nations that implement specific characteristics, i.e., (1) the establishment of regulations that govern enterprise behavior toward recycling EoL batteries; (2) the imposition of hefty fines for any violations; (3) the regulation of recycling processing fees, taxes, and deposit payments, which serves to oversee recycling enterprises and enforce consumer recycling obligations. However, in China, the recycling of rechargeable batteries is encouraged by charging consumers a fee for recycling treatment and providing subsidies. As a result, China has an imperfect and ineffective recycling system in comparison to several other countries, and most EoL power batteries have not been recycled through formal methods [53]. Countries with low recycling rates typically lack well-developed recycling systems and measures to monitor enterprises [29].(2)Currently, the policies of several countries prioritize the recycling and echelon utilization of power batteries, which are important for implementing sustainable development. The field of power battery echelon utilization is still in its early stages, and the policies introduced by various countries primarily focus on funding the R&D of enterprises in this field. The application scenarios for echelon utilization are rather vague, and there has yet to be a comprehensive and specific summary of them.

## 3. Existing Recycling Models

EoL power batteries are associated with strong negative externalities [54]. Two main reasons contribute to this. Firstly, consumers dispose of EoL batteries via inappropriate recycling methods. Secondly, enterprise battery reuse technology is often outdated, subsidy funds are limited, and recycling subject rights and responsibilities are unclear [41]. As a result, the process of recycling EoL power batteries can be divided into recycling methods, and systems. The recycling mode refers to how enterprises retrieve a large number of EoL power batteries from consumers. The recycling system outlines the principles and methods of handling batteries once enterprises collect the EoL power batteries. Therefore, the development of an efficient recycling method and system is essential for effectively recycling EoL power batteries.

### 3.1. Existing Recycling Methods

#### 3.1.1. Battery Manufacturer’s Recycling Model

The battery manufacturer recycling model involves battery manufacturers collecting EoL batteries from consumers through methods such as NEV manufacturers, NEV retailers, and EoL NEVs dismantling companies, establishing a recycling method led by battery manufacturers and based on reverse logistics [55]. In recycling network construction, manufacturers can build their own recycling network to complete recycling, or they can cooperate with other companies in the supply chain and use the forward logistics network to transform into a reverse recycling network to form a closed loop of recycling. There are three challenges to this recycling model. First, this model requires the company to work with upstream and downstream companies to transport the product from the recycling to the manufacturing end. If there are problems in the recycling process, it is difficult for battery manufacturers to recover many EoL products from consumers. Second, recycling companies need much capital to invest in the core technology of the whole industry and it is challenging to generate scale effects for profit in the short term. Third, logistics costs for recycling companies to transport EoL products over long distances are high, accounting for approximately 70% of the total recycling costs. Therefore, this type of recycling requires upstream and downstream companies to work together to recycle or requires government subsidies and support to complete the recycling process successfully [56,57]. The recycling process is shown in Figure 4.

Japan has established a recycling method mainly led by battery enterprises, with the idea of “reverse logistics”. Battery manufacturers use service networks such as retailers, sellers, and gas stations to recover EoL batteries from consumers free of charge and then hand them over to professional battery recycling companies for disposal. Toyota Motor Corporation offers customers cash subsidies or purchase discounts for returning EoL batteries to dealers or retailers. The Nissan Leaf once offered a battery replacement business for customers [58].

#### 3.1.2. NEV Enterprises Recycling Model

Under the NEV enterprises recycling model, NEV enterprises will be responsible for recycling (EoL) vehicles. NEV enterprises can use their existing sales distribution network to establish recycling service stations, such as automobile sales, 4S stores, and after-sales service stations, or they can establish specialized recycling service stations. NEV enterprises recycle power batteries and transfer the recovered batteries to regular dismantling enterprises. The batteries undergo echelon utilization and resource recovery. The recycling process is shown in Figure 5.

BYD is the first company in China to lay out the NEV market. BYD’s recycling method for EoL power batteries is mainly through authorized retailers or dealers. When a customer needs to replace the EoL power battery, the retailer or dealer will transport the EoL power battery to the BYD factory for testing. Trumpchi, FAW-Volkswagen, and Geely are establishing special recycling service stations for recycling. The three companies have established the largest number of recycling service stations in China, accounting for 26.9%. In addition, many NEV manufacturers actively sign strategic cooperation agreements with battery recycling and dismantling enterprises to jointly establish power battery recycling networks. According to data from the MIIT, as of November 2022, China has established 15,251 power battery recycling service stations nationwide.

#### 3.1.3. Industry Alliance Recycling Model

The industry alliance recycling model refers to an incentive alliance between manufacturers, sellers, recycling, and traceability technology enterprises. In the recycling process, the sales and service networks among the members combine to build a closed-loop recycling network to centralize the recycling of EoL power batteries. The EoL power batteries are transported to the professional recycling centers of the alliance members for recycling and processing. Compared with individual manufacturers or retailers, industry alliances can divide and collaborate according to their strengths, ensuring the integrity of the entire recycling industry chain. The specialized division of labor within the industry alliance can improve recycling efficiency, allowing the entire recycling system to have a more professional recycling network and processing equipment at a lower cost [57]. The recycling process is shown in Figure 6.

The industry alliance recycling model is popular in the U.S. The ‘Portables’ team in the U.S. is an incentive consortium that was formed by Everledger, HP, Fairphone, and Call2Recycle. Everledger provides technical support for the battery traceability feature, which creates a battery passport for manufactured batteries. HP and Fairphone are the manufacturers of lithium batteries, and Call2Recycle is the company responsible for recycling lithium batteries, including EoL power batteries [29]. Call2Recycle operates 136,000 battery recycling stations throughout the U.S.; 86% of Americans live within 10 miles of a recycling facility. Furthermore, the program recycled more than 8.4 million pounds of used batteries in 2020, including nearly 2.5 million pounds of lithium batteries that would otherwise have been disposed of in landfills [59]. The U.S. collected more than 8.1 million pounds of used lithium-ion batteries for recycling in 2021, an increase of 12.3% over 2021 [60]. The Portables team has the most extensive management, collection, logistics, and recycling program for batteries in the U.S. The key to promoting the sustainable development of (EoL) power batteries is the industry consortium’s recycling model, which involves government, upstream, and downstream industry chain companies, as well as consumer cooperation.

#### 3.1.4. Third-Party Recycling Model

Under the third-party recycling model, manufacturers pay a service fee to third-party companies so that third-party companies can assist companies in completing the EPR. The service fee is calculated based on the quantity and type of EoL power batteries returned by the manufacturers. This model can transfer the responsibility and risk of recycling to the third-party company, so it is more suitable for small manufacturers [61]. Third-party companies need to independently build recycling networks and collect EoL power batteries through various recycling methods, which reduces the difficulty of recycling. After collecting a large number of EoL power batteries through the recycling network, the third-party company will transport the battery back to the recycling and processing center and realize large-scale production with the technical advantages [52]. Recycling and processing centers will have valuable metal materials that can be resold to battery manufacturers. The critical issue in the recycling model of third-party companies is the company’s authority. Therefore, the state should strictly manage the accreditation of third-party companies. The recycling process is shown in Figure 6.

In China, GEM is a third-party recycling company approved by China. As a leading company in recycling EoL power batteries, GEM built a full life cycle value chain from recycling, remanufacturing, and echelon utilization to ensure maximum reuse of EoL power batteries. Moreover, MAE built an “Internet+recycling” platform for EoL lithium batteries by cooperating with battery manufacturers, NEV manufacturers, EoL battery recycling and dismantling companies, automobile retailers, and internet e-commerce platforms. This innovative recycling model has built a nationwide recycling service network and realized the efficient recycling of metal resources (refer to Figure 7).

### 3.2. Existing Recycling Systems

A low-carbon, resource-efficient, and inclusive future requires CE systems that emphasize reducing, reusing, and recycling [62]. CE systems can reduce the amount of final waste, reduce the use of virgin natural resources, and increase product efficiency, thereby improving environmental health [63]. Liao et al. [64] proposed that the closed-loop supply chain should comply with the 3R system, i.e., reusing, remanufacturing, and recycling. However, no studies have summarized the EoL power battery recycling system. Therefore, this paper categorizes the EoL power battery recycling system into four aspects—reusing, remanufacturing, recycling, and reducing.

#### 3.2.1. Reducing

Reducing means that society uses as few non-renewable and natural resources as possible in the process of production and consumption, reduces the amount of energy used, and improves the utilization of resources; it is also a prerequisite and basis for recycling and resourcefulness [65]. Reducing can be implemented in three dimensions: at the source, in the middle, and end. Reducing the source of producing remanufactured batteries reduces the re-mining of non-renewable metals, enables the re-recycling of resources, and improves utilization rates. The reduction can be reflected in two aspects of the production of power batteries. On the one hand, using green and recyclable materials, manufacturing processes, and recycling technologies should be considered in the manufacturing and design of power batteries, thereby reducing the use of materials and energy consumption. This will reduce the use of materials and energy consumption, save resources, and reduce pollution at the source. On the other hand, to address the scarcity of metal resources and reduce the environmental damage caused by mining metal resources, power battery manufacturers reuse recycled metal resources to reduce the use of metal materials from the source of production.

#### 3.2.2. Reuse

After a battery is repaired (after its first use), it can be reused in the same vehicle model. This method reduces the need for new batteries to be produced for use in vehicles [66]. The battery can still be disassembled and, after simple treatment, be applied to the same make of NEV in exceptional cases, such as when a NEV breaks down or experiences a car accident during use, resulting in the vehicle being scrapped. The remaining capacity of the built-in power battery is still greater than 80% of its original capacity in such cases. During the process of recycling EoL batteries, reliability and compatibility are important considerations [67].

#### 3.2.3. Remanufacturing

Essentially, remanufacturing is the process of repairing, replacing, or restoring EoL products to ‘like-new’ conditions [68]. This refers to the process of transforming EoL power batteries so that they meet the standards of existing power battery standards and can be reused. Compared to traditional manufacturing, remanufacturing represents an advanced form of a CE that can significantly reduce the use of raw materials and extend the life of products. The remanufacturer collects a certain amount of EoL products from recyclers, and the recycling department remanufactures them based on their service life and degree of damage [69]. The remanufacturing process typically involves complete battery testing, partial disassembly and replacement, and the reassembly of the battery pack. Generally, small modules in the cell are responsible for degrading a power cell’s performance [68]. In order to complete the remanufacturing process, the faulty cells or modules are identified and replaced with qualified components. Batteries remanufactured are then sold to automotive OEMs or the spare parts market [70]. A remanufactured part can meet or even exceed the quality and performance of a new part, saving costs, reducing energy consumption, and reducing carbon emissions [71,72]. Although remanufacturing can reduce the cost of batteries by 40% compared to brand new batteries, this method is not widely used. In practice, laddering is used in a more significant number of situations [73].

#### 3.2.4. Recycling

Recycling companies have become the most popular method for recovering rare metals and valuable materials from power batteries [31,74]. By adopting this approach, the battery industry can reduce its environmental footprint, waste volume, and dependence on imports of crucial metal resources [32]. Battery manufacturers, vehicle manufacturers, and third-party recyclers can all recycle power batteries [67]. Generally, the anode of a power battery is composed of graphite, while the cathode is mainly composed of NMC, LFP, and NCA [73]. The most commonly used recycling technologies for lithium-ion battery packs are pyrometallurgy, hydrometallurgy, and direct recycling, each with its own advantages and disadvantages. Globally, battery recycling technologies and methods are relatively mature, and many countries use wet and pyrometallurgical methods to recycle lithium batteries [48]. Through the above-mentioned technical routes, valuable metal resources can be effectively recycled at the source.

In conclusion, each system places much emphasis on reusing. However, the existing studies do not specify the process of reusing. In the EoL power battery recycling process, we specify that the process of echelon utilization involves reusing. From the policy analysis in the previous section, it is clear that echelon utilization is already a key area for future research and development.

## 4. Echelon Utilization

Echelon utilization can be defined as reusing the EoL power batteries. Echelon utilization can relieve the pressure of recovery, reduce environmental pollution, improve economic efficiency, and help the development of renewable energy. Echelon utilization aims to reduce the battery capacity to less than 80%, which is a slight scrap, and is not suitable for power batteries used in NEVs [75]. EoL power batteries should be reused in echelon utilization before being recycled. Directly recycling and scrapping the EoL power batteries with 40–80% leads to a serious waste of resources. Generally, there are four steps to the echelon utilization of EoL power batteries—disassembly, residual energy detection, screening, and recombination [43,76]. Detecting and screening out batteries with different loss levels is crucial.

### 4.1. Application Scenarios

As the reserve energy of EoL batteries can be applied to some scenarios, it is extremely critical to pay more attention to the echelon utilization. There are three categories of application scenarios for echelon utilization: generation-side energy storage, user-side energy storage, and mobile power supply. The classification of echelon utilization scenarios presented in this paper is based on the classification of energy storage proposed by [77] as well as the theoretical model of super-capacitors proposed by [78].

In general, the division of power generation-side and user-side energy storage systems is based on the size of energy storage facilities and the standard of centralized physical distribution of energy storage systems. The scale of the generation-side energy storage is usually large and the level of battery consistency and balance management is high. ThU.S., it often needs the support of a large power grid [79], such as renewable energy collection, peak-load shifting, and voltage or reactive power support. The scale of user-side energy storage is relatively small and the consistency requirement for batteries is low. For these reasons, distributed energy storage is relatively popular at the moment. A wide range of EoL batteries are available, which can be used in a wide range of scenarios in urban life, the construction of microgrids in rural areas, and the improvement of basic power facilities on farms. In addition to recycling EoL batteries, there are a number of companies that support the development of user-side energy storage applications. It is critical to note that mobile power supply is essentially a form of user-side energy storage. This is due to the fact that stationary battery systems differ from vehicle battery systems in certain respects, so we have classified it as a third category in order to be able to clearly describe specific application scenarios. The classification of the three echelon utilization scenarios is shown in Figure 8.

#### 4.1.1. Generation-Side Energy Storage

There are two main application scenarios of energy storage cited in this paper (the storage of renewable energy and “peak cutting and valley filling”) to ensure the stability of the power grid on the generation side.

As an energy storage facility, batteries are primarily used for storing sustainable renewable energy [79]. Applications of renewable energy include the storage of solar energy and wind energy. Batteries in NEVs have become cost-effective battery raw materials that can be incorporated into the power grid as large-scale battery energy storage systems [80]. With the advancements made in science and technology, the contribution of electricity generated from this type of energy is increasing. The intermittency of this energy supply calls for a step change in energy storage technology. Generally, [78] suggested that batteries are considered a type of technology that has high-energy–low-powered density. The ultra-capacitor is an energy storage device with both high- and low-powered density. It is possible to use the combined ultra-capacitor to supplement batteries and provide pulsed cycle storage for hybrid energy storage by bridging the gap in energy density between batteries and ordinary capacitors. Hou et al. [81] specifically discussed the combination of lithium-ion batteries and supercapacitors. Many companies have already developed applications in these areas. The Bosch Group has begun to recycle batteries. The company acquired key battery recycling companies. In 2015, it successfully constructed a large-scale photovoltaic (battery storage) grid system utilizing used batteries from BMW ActiveE and i3 pure NEVs [82]. In Bangladesh, 200,000 solar home systems are using second-hand lead–acid batteries [83]. Solar energy is stored and supplied using battery pack systems. In 2011, GM and ABB (one of the world’s largest power and automation technology companies) developed a smart grid that used batteries from used GM electric cars and Chevrolet Volt battery packs. The system was designed to store wind and solar energy and feed it into the power grid [34]. In 2017, Mercedes-Benz built energy storage stations using EoL batteries from 1000 NEVs [84].

Using batteries as energy storage systems on the generation side can also cut and fill peaks and valleys. Due to the greater relative cost of energy consumption during peak periods than during low peak periods, the battery can be used as an energy storage device at a low cost to store excess energy during peak periods and then output the stored energy during low peak periods to the power grid. Generally, the application scenarios for energy storage on the power generation side have high demands on battery specifications, packaging forms, and quantity. In addition, they also have high demands on the level of selection and balance of management of batteries. For example, GM and ABB are further exploring the use of used battery packs to provide backup power for small commercial buildings when power is lost as a means of compensating for the intermittent power generation limitations of renewable energy sources, such as solar and wind power, as well as storing and using electricity during preferred periods in a way known as “peak-load shifting”.

#### 4.1.2. User-Side Energy Storage

The main characteristics of the application scenarios of user-side energy storage of EoL batteries involve distributed energy storage, such as small charging stations, commercial power supply, remote areas without power grids, and scattered villages with high construction costs for power grids.

In economics, the EoL battery industry is still in its infancy, and there is less risk associated with small applications. The current application of user-side energy storage has greater economic potential than power generation-side energy storage. In this area, some companies have already begun implementing plans. For example, Japanese automaker Nissan has pre-ordered a residential energy storage unit called “xstorage”, which uses 12 retired EV batteries and is connected to the grid to provide backup energy [80]. To demonstrate the feasibility of reusing EoL batteries as a source of home energy supply, Duke Energy and Tokyo Itochu are working together to explore the possibility of diversifying EoL batteries for energy storage devices. Microgrids and distributed new energy can be used as EoL batteries in a limited range of urban and rural applications, and echelon utilization of EoL batteries has obvious advantages over new batteries. In a recent article, [83] reported that EV batteries could power homes and off-grid facilities supporting healthcare facilities, telecommunication towers, businesses, and other high-capacity users. China Tower successfully established a decommissioned energy storage project of power batteries in Henan Province in August 2013, which was also the first solid hybrid microgrid system based on a decommissioned power battery in China. As of the end of 2018, China Tower Corporation started using EoL power batteries in about 120,000 base stations across 31 provinces in China, which are widely used in standby electric energy storage devices. Heymans et al. [85] summarized some application scenarios of user-side energy storage primarily in urban scenarios, including residential telecommunication towers, light commercial, office buildings, fresh food distribution centers, and transmission support, as well as provided detailed information on the potential and limitations of use in these scenarios. In rural energy storage applications, EoL batteries are applicable. In rural areas, rural electricity grids are some of the most important infrastructure types, and lead–acid battery packs are the most common forms of energy storage. Due to the short lifespan of lead–acid battery packs, the cost of energy storage and carbon emissions are greatly increased [86]. Practical production requires reducing battery costs, and the costs of used EV batteries are considerably lower than those of lead–acid batteries [86]. For this reason, using retired batteries from NEVs to constructing an independent microgrid in rural areas is very important for improving household power systems. Irrigation systems are a good example of how retired batteries can be applied in rural areas. The project is based on the principle of ‘one well and one circuit, one meter and multiple cards’ [82]. Some wells are located in areas that are distant from distribution rooms. Building distribution rooms and power circuits for these scattered wells is costly, resulting in low utilization and high waste. With the advent of lithium-ion battery-powered irrigation pumps, these dispersed Wells can be monitored and controlled without needing electrical connections. Furthermore, a used lithium-ion battery irrigation pump is flexible and easy to maintain. The investment cost is also very low, bringing great benefits to rural farmers. Similarly, retired lithium-ion batteries are used to illuminate more isolated country roads.

#### 4.1.3. Mobile Power Supply

Mobile power supply mainly refers to the echelon utilization of used batteries in low-speed vehicles and electric motorcycles that have less battery performance requirements than NEVs. Applications include power batteries for vehicles with low-performance requirements, backup power supplies for short trips in NEVs, and emergency start energy. Jin [82] demonstrated that recycled batteries perform better than traditional lead–acid batteries when used in electric forklifts and low-speed electric utility vehicles. They also have obvious economic advantages. In addition to low-speed EVs, EoL NEV batteries are also suitable for electric motorcycles. Due to the lower performance requirements of electric motorcycles, the lithium-ion batteries of used EVs can continue to be used as the power source for electric motorcycles. With the rapid growth in demand and quality of EoL batteries, third-party enterprises are expected to become one of the main forces in power battery recycling [87].

### 4.2. Development Status and Trend

Echelon utilization was first studied in Europe, the U.S., and Japan. Currently, chemical batteries, fuel batteries, and solar batteries make up the majority of power batteries on the market. Chemical batteries include lead–acid batteries and lithium-ion batteries. Lead–acid batteries are the most widely used in the automotive industry due to their low price and mature technology [88]. There has been a gradual increase in the use of EoL lithium-ion batteries throughout the world in recent years. It is anticipated that EoL lithium-ion batteries will become an essential component of national distributed energy storage systems due to their ability to extend battery life. Each energy storage scenario has its advantages. Implementing generation-side energy storage is more profitable and environmentally beneficial, while it is less risky to implement user-side energy storage.

Rapid echelon utilization development has led to the significant strengthening of cooperation among leading enterprises within the foreseen scope, with the strong support of national policies [79]. The government has devised a series of incentives and policies to encourage the industry to invest in and build large-scale projects in line with the national strategy. At present, many laboratories and enterprises around the world are collaborating with leading NEV manufacturers to evaluate the EoL batteries of their products in various applications. For example, China tower, the China Electric Power Research Institute, State Grid Electric Power Company, Pacific Northwest National Laboratory, the National Renewable Energy Laboratory, and Duke Energy. ThU.S., scientific research institutions, power battery manufacturers, and energy storage integrators learn from the experience and practices of advanced regions to promote the construction of recycling systems. Industrial technology innovation has become an important trend for future development. Multiple institutions are cooperating and exploring diversified business models and building support platforms to improve economic and environmental sustainability.

## 5. Proposed Trends in EoL Power Battery Recycling System from a Circular Economy Perspective

### 5.1. Theoretical Basis of the Recycle System

EoL power batteries refer to three types of situations. First, EoL batteries are created during the production process by battery manufacturers. Second, NEVs cannot guarantee regular operation due to the remaining capacity or the charge/discharge performance. Thirdly, EoL power batteries have been reused after echelon utilization. The ability to recycle power batteries effectively will directly impact national energy savings and emission reduction strategies. Integrating the battery recycling production network into the industrial ecological chain will optimize the allocation of system resources and minimize external diseconomies. However, we found that there currently needs to be clearer guidance for recycling EoL power batteries. Based on the analysis in Section 3 and Section 4, this section proposes a specific recycling pathway to provide a clear approach for government policies and enterprise implementation, which have strong theoretical bases and practical significance.

In general, recycling EoL power batteries can be divided into two parts: recycling methods and recycling systems. Countries around the world can determine a reasonable recycling paths according to the country’s development and combine it with the four recycling methods proposed in Section 3.1. As for the recycling system, researchers have yet to propose a system for recycling EoL power batteries. According to the existing policy base, CE theory, and closed-loop supply chain system, the current overall industrial development trend is gradually focusing on the stage of echelon utilization. Therefore, based on the existing 3R system, this paper proposes a 4R recycling system that considers echelon utilization, including reusing, remanufacturing, recycling, and reducing. In addition, there is no clear indicator for different recycling steps to judge the state of the battery. Researchers have defined battery health status based on the battery’s capacity, internal resistance, and power. However, these concepts have relatively narrow definitions and analyses. They cannot be used to determine whether a battery with EoL power remains healthy throughout its full life cycle [40]. Monitoring and predicting the battery’s state of health (SOH) is extremely important for the future intelligent battery management system (BMS) [89]. Intensive research has been conducted on evaluating SOH in commercial power batteries [90]. Therefore, we chose the SOH to assess the state to determine the specific steps for recycling.

### 5.2. 4R EoL Power Battery Recycling System

In the early stages of developing NEV batteries, the open-loop production mode of “cradle to grave” was mainly utilized. The industrial ecological chain ends at the product’s EoL point. This paper presents a 4R recycling system for EoL batteries. In this paper, we aimed to design a 4R system for recycling EoL power batteries, addressing the industry’s original crude open-loop production structure and achieving a “cradle-to-recycle” production model [91,92].

In Figure 9, the blue line indicates reverse logistics and the red line indicates forward logistics. In order to manufacture power cells, raw materials must first be extracted from the soil for the elements or imported from abroad. By extracting and processing raw materials, power cells can be manufactured. Based on the performance and characteristics of each cell, the elements are combined into a battery pack. In addition to the battery module, the battery management system (BMS) and the connection components are included in the power pack. The battery pack is sent to a battery manufacturer to produce the power cells, and the battery pack is then sent to a battery manufacturer to produce the power cells. The battery manufacturer then sends the power cells to the NEV manufacturer. The production of NEVs is then completed by the companies that sell the vehicles to consumers.

After 5–8 years of consumer use, the power battery will be retired. Consumers take their EoL power batteries to a standard recycling service station and receive a subsidy offer. The recycling service station hands the EoL power battery to the NEV manufacturer. NEV manufacturers fulfill the EPR by returning EoL power batteries to the upstream battery manufacturer. Prior to battery echelon utilization, the battery manufacturer calculates the SOH of the battery through BMS. The enterprise can assess the economy, safety, and availability of power batteries by reasonably evaluating the SOH data of vehicles and assessing the application field of the secondary reuse of power batteries [93,94,95]. When SOH is expressed as a percentage, a value of 0 indicates that the battery is empty; a value of 1 indicates that the battery is full. There are two main types of power battery recycling: echelon reuse and resource recovery. A SOH value between 0.8 and 1 indicates that the battery is good quality and can be reused directly after remanufacturing. In accordance with China’s requirements, echelon utilization scenarios can be used when the SOH value is between 0.4 and 0.8. When the SOH value is between 0 and 0.4, the battery cannot be used, and the metal elements are removed for recycling [48,96]. Lithium, nickel, cobalt, and copper can be recovered using this process.

The 4R recycling system for EoL power batteries improves the components’ utilization rate and reduces the industry’s ecological disorder. Power batteries can be extended in life, and a “produce-use-recycle-reuse” production model can be used and enterprises will be able to implement a “cradle-to-cradle” production model for power batteries. Meanwhile, it creates several market opportunities in the recycling system, reducing waste and negative externalities on the environment. If the government and enterprises actively implement the 4R recycling system in the future, it will be of great importance to improve the country’s economic, environmental, and social well-being.

## 6. Existing Problems and Recommendations

### 6.1. Existing Problems

Previous studies have shown that recycling power batteries is effective, i.e., economically, environmentally, technically, and resourcefully [13,25,74]. However, there are still many things that could be improved in the development of the recycling power battery industry. Based on the six aspects of industrial development, this section analyzes the possible impediments to the future industry from six aspects: environment, technology, facilities, economy, policy, and society.

#### Existing Policy Problems

Government policy plays a critical role in the emerging industry, but five issues remain regarding policies. (1) The NEV battery recycling industry needs a transparent system and specific recycling and dismantling standards. NEV manufacturers consider factors such as safety, space optimization, and applicability during the design process. However, there are differences in the battery packs, modules and battery packs produced according to the requirements of different vehicle models [47]. It will create specific difficulties for further dismantling by battery recycling companies. (2) The recycling rate set by the government is high. Using China’s 2019 data as an example, the recycling rate of EoL power batteries in China is 24.8%, which shows that the recycling rate is much lower than the initial target set [52,96]. (3) The government still needs to clarify the responsibilities of enterprises at each stage of the whole life cycle of power batteries [97]. There are no concrete punitive measures for enterprises that do not strictly implement the standards. When batteries enter the second life stage, many local governments must clarify the primary responsibility for strict recycling. At the same time, the government has a moderate punishment for enterprises that fail to fulfill their responsibilities, making the recycling system difficult to achieve. (4) Most government policies focus on the enterprise side, needing more proper guidance and consumer incentives. From the current policies introduced by some local governments, most government subsidies focus on subsidies for recycling enterprises, but ignore consumer incentives. The lack of strong government incentives for consumers can lead to a lack of interest in remanufactured and recycled EoL products. Consumers may sell their EoL power batteries to small workshops for more significant financial gain. At the same time, consumers are psychologically skeptical of the quality of products being remanufactured, which can make it difficult for EoL batteries to return to the market [98]. (5) The government and enterprises neglect advertising to consumers. Advertising not only stimulates consumer demand but also plays a vital role in increasing consumers’ willingness to recycle EoL power batteries [99]. Currently, most EV advertisements focus on increasing consumer loyalty, promoting environmental protection and acceptance of new products. Notably, they often need to pay more attention to promoting leadership and publicity regarding channels and methods for collecting and recycling EoL power batteries [100]. Consumers only notice that EoL power batteries can be recycled and remanufactured after receiving promotional messages or trade-in policies [101]. (6) In some countries, implementing EoL power battery recycling may be limited by financial and economic barriers. In order to recover large volumes of EoL power batteries, governments must establish multiple infrastructures, such as collection, sorting, disassembly, recycling, and remanufacturing facilities [56]. As these centers require high-tech and highly skilled labor, the government must invest substantially. Therefore, this could challenge countries with high labor costs, such as the EU. The feasibility of completing this project is challenging to assess [102].

### 6.2. Existing Technical Challenges

Recycling EoL power batteries may have some negative externalities on the environment [103]. We analyze the current development problems from the technical point of view for the following reasons. (1) Metal mining has negative externalities for local communities, polluting air, soil, and water resources and causing adverse health effects for the population [22,104]. (2) Using conventional waste disposal methods (e.g., landfills and incineration to dispose of EoL power batteries) will also seriously pollute the environment. When EoL power batteries are disposed of in a landfill, the positive shell is susceptible to external damage and the internal components are exposed to leachate [74]. If these contaminants are not properly disposed of, they could threaten human health and the environment [105]. (3) There are still negative impacts associated with reverse logistics recycling activities, as recycling, sorting, remanufacturing, disassembly, and recovery require large amounts of energy and significantly impact the environment [106,107]. However, even though the total energy consumption of lithium-ion battery recycling should be less than that of battery production at the start of the supply chain, the recycling system still consumes a large amount of energy. Recovering and recycling waste requires using chemicals, heat treatment, and machinery. Another challenge in this area is the complexity of measuring and monitoring environmental practices [108]. Energy and chemicals are consumed in dismantling, acid leaching, chemical precipitation, and regeneration of batteries. As a result of the use of these chemicals, secondary waste can be generated as well as greenhouse gas emissions. (4) Companies cannot meet the standards proposed by the government in advance due to technical limitations in the recycling system. Many countries have set higher standards for the recovery rate of valuable metals. In the published Chinese documents, the total recovery rate for nickel, cobalt and manganese must not be less than 98%, for lithium not less than 85% and other primary valuable metals such as rare earth not less than 97% [51]. According to EU regulations, recovery rates for cobalt (90%), nickel (90%) and lithium (35%) are difficult to achieve in practice [30]. The lack of proper tracking technology in the power battery recycling system and possible uncertainties in the quality and quantity of EoL batteries in reverse logistics further add to the barriers to recycling [47]. (5) Recycling facilities for small businesses are limited by their recycling capacity, costs, and resources, making it difficult to achieve economies of scale [109]. There may be increased costs and a reduction in profits for companies, leaving them with no incentive to recycle [102,110].

### 6.3. Recommendations

Due to the rising costs of raw materials for power batteries and graphite, recycling (EoL) power batteries has gained worldwide attention. In this paper, we analyze the recycling policies of representative nations and find that nations worldwide are gradually focusing on the scenario of hierarchical utilization of EoL power batteries. Based on the current recycling model and system, we propose that the 4R recycling system can effectively manage EoL power batteries in the future. Regarding the six presented development obstacles, we propose the development of recycling EoL power batteries from the perspectives of government, enterprises, and consumers.

#### 6.3.1. Government

(1)According to the 4R EoL power battery recycling system, governments consider echelon utilization and clarify the responsibilities of recycling enterprises. To achieve the goals of environmental protection, resource conservation, and safe use, the government must also improve its policies and regulations, technical guidelines, evaluation standards, and industrial chain for echelon utilization.(2)The government should bolster oversight of EPR implementation by battery manufacturers and NEV manufacturers and improve the regulatory framework. In addition, the government should increase the penalties for companies that violate recycling laws.(3)The government establishes technology subsidies, focusing on subsidizing research and technological breakthroughs in critical echelon utilization applications. The government has increased support for recycling companies to promote the growth of the recycling power battery industry.(4)The government increased subsidies to consumers to incentivize consumer participation in recycling. The government and competitors should educate consumers about environmental awareness and formal recycling procedures.(5)The government rationalizes the planning of recycling infrastructures for power batteries. It may be convenient for consumers to return their power batteries at these locations and for enterprises to reduce transportation distances and reduce costs by replacing them at these locations.

#### 6.3.2. Enterprises

(1)Battery manufacturers, NEV manufacturers, NEV retailers, and hierarchical utilization enterprises must engage in collaborative design, management, and data sharing, and cooperate to address the issues of constructing battery recycling methods, building a recycling evaluation system for the entire life cycle, and establishing a service system. According to the four existing recycling methods, technology integration among enterprises will be strengthened, thus reducing investment risks and achieving win-win cooperation.(2)Battery manufacturers should increase their investments in recycling equipment, research, development, and technical personnel. They should establish a traceable power battery recycling regulatory system using blockchain technology to ensure real-time monitoring of companies’ implementation of full life cycle tracking. Enterprises should also enhance their awareness of recycling and improve their sense of corporate social responsibility.(3)NEV retailers should use advertising and education to increase consumer participation in battery recycling or promote “trade-in” activities to attract more consumers to participate in recycling. When the recycling volume reaches a particular scale, it can also increase the profit of the recycler. In addition, recycling companies should set reasonable prices to balance the interests of enterprises and consumers.

#### 6.3.3. Consumers

As essential participants in recycling EoL power batteries, consumers should recognize the critical role of environmental protection and resource conservation. Consumers should consciously comply with their recycling obligations and take the initiative to send EoL batteries to designated recycling service stations. By consciously recycling, consumers can gain considerable benefits and effectively reduce the costs of government regulation, thus achieving the goal of maximizing social welfare.

## 7. Conclusions

With the rapid popularity of NEVs, large-scale EoL power batteries will face retirement. There are substantial potential development opportunities and broad market prospects for recycling EoL power batteries. In order to reduce environmental damage, save resources, and protect health, it is important for the government, enterprises, and consumers to remanufacture, utilize them hierarchically, and recover resources from EoL power batteries through cooperation under a circular economy. This paper compares the current industrial development status and analyzes the problems of the existing recycling process in terms of policy and technology from a practical problem perspective. The main conclusions of this paper are summarized as follows:(1)This paper examines the battery recycling regulations of the EU, the U.S., Japan, and China. The government regulates countries with high recycling rates through legislation, deposit systems, high fines, and subsidies. Upon analyzing the policies, it was found that echelon utilization of EoL power batteries has become a significant concern in all nations.(2)This paper summarizes the four existing recycling “models”: battery manufacturers recycling model, NEV enterprises recycling model, the industry alliance recycling model, and the third-party recycling model. This paper summarizes the recycling systems in EoL power batteries, including reducing, reusing, remanufacturing, and recycling, by combining circular economy theory and closed-loop supply chain systems.(3)Combined with the development policy and recycling system, EoL power battery echelon utilization will be the focus of future development. However, several studies clearly delineate the application scenarios of echelon utilization. This paper presents the application scenarios of echelon utilization in terms of generation-side energy storage, user-side energy storage, and mobile power supply; we use actual cases for analysis.(4)The government must establish clear and effective recycling methods and normative recycling systems. Therefore, this paper proposes a 4R EoL power battery recycling system that accounts for echelon utilization and suggests the use of state of health (SOH) to assess the states to determine the recycling steps.(5)This paper analyzes the existing policy problems and technological challenges and proposes three aspects of development for the government, businesses, and consumers based on the above analysis.

This paper has a limitation in that it mainly summarizes the literature studies and actual situations without using bibliometric analysis, which is commonly used by most scholars. Therefore, future studies can use this approach to analyze the literature in this field from multiple perspectives.

## Figures and Tables

**Figure 1 ijerph-20-04346-f001:**
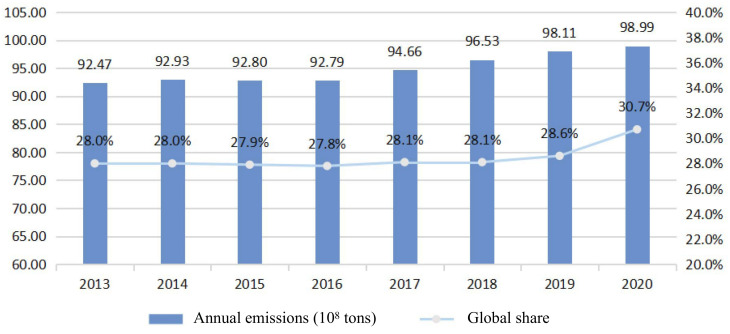
Development trend of carbon emission in China in recent years (2013–2020).

**Figure 2 ijerph-20-04346-f002:**
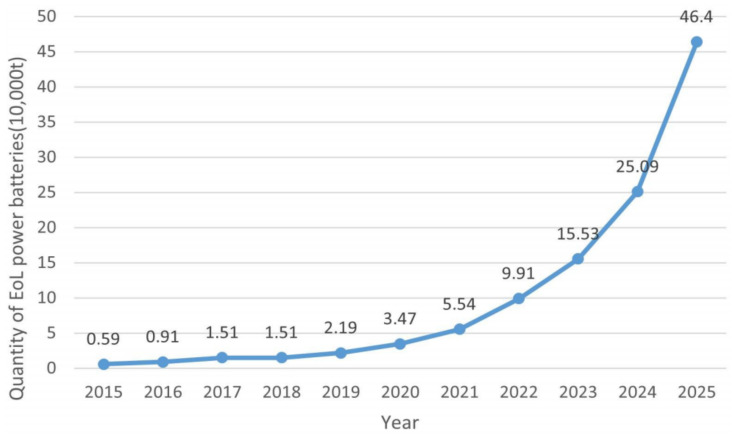
Forecast of the number of EoL power batteries in China by 2050 [18].

**Figure 3 ijerph-20-04346-f003:**
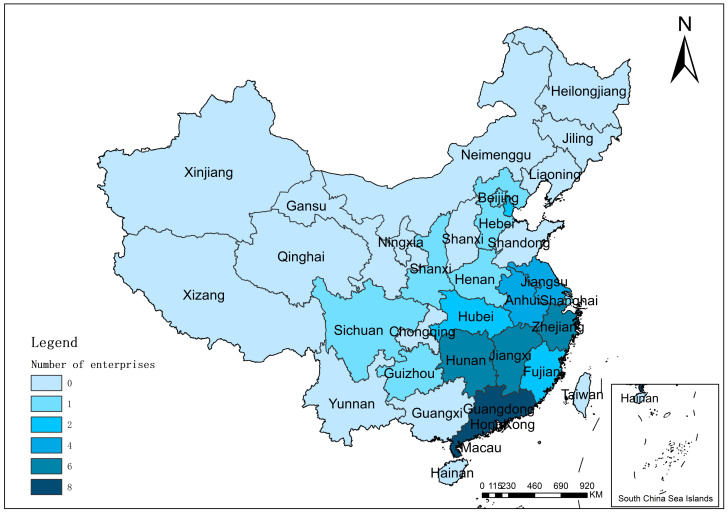
Location of whitelisting power battery enterprises in China.

**Figure 4 ijerph-20-04346-f004:**
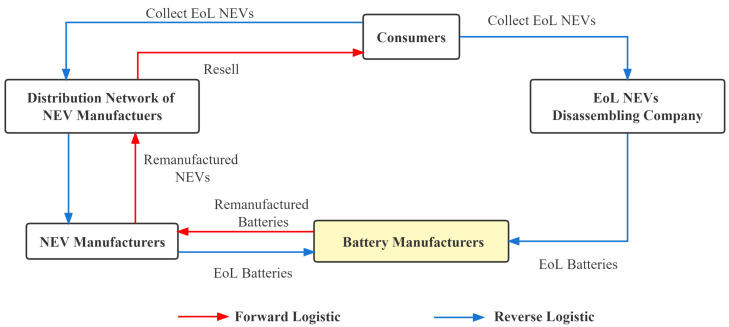
Battery manufacturer recycling model.

**Figure 5 ijerph-20-04346-f005:**
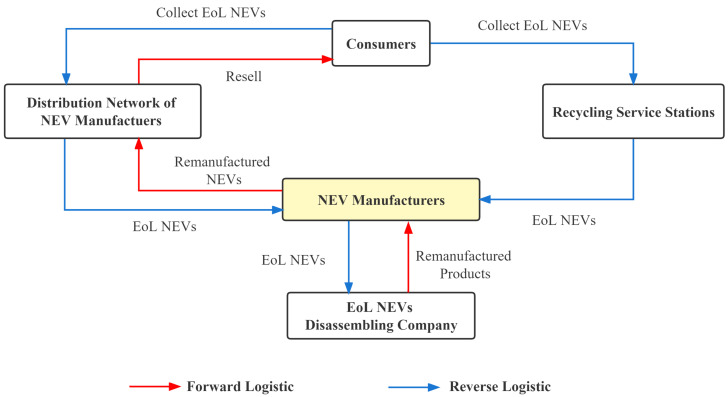
NEV enterprise recycling model.

**Figure 6 ijerph-20-04346-f006:**
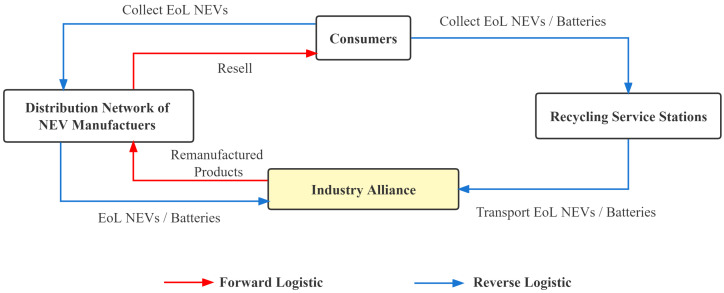
Industry alliance recycling model.

**Figure 7 ijerph-20-04346-f007:**
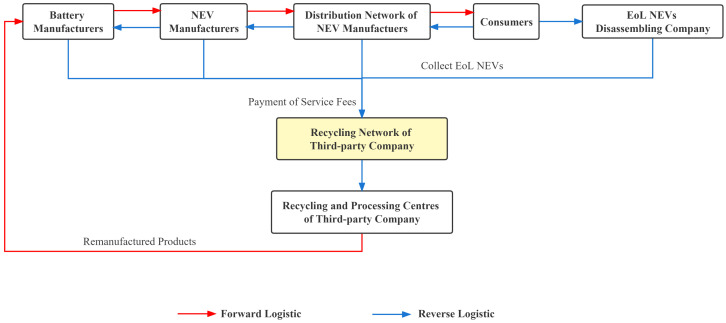
Third-party recycling model.

**Figure 8 ijerph-20-04346-f008:**
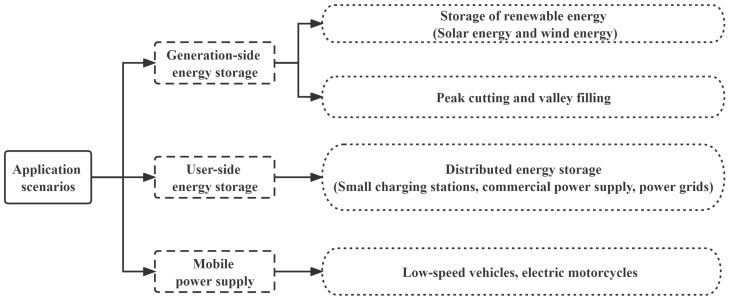
Application scenarios for echelon utilization.

**Figure 9 ijerph-20-04346-f009:**
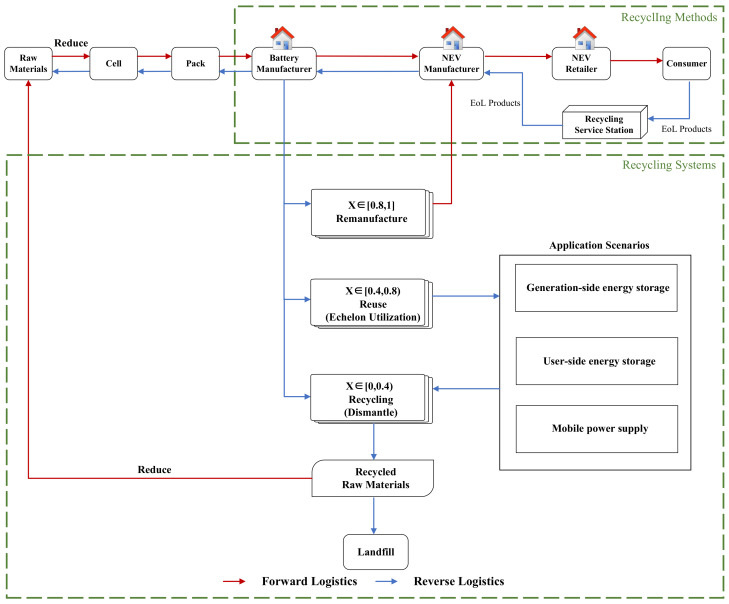
Flowchart of the 4R EoL power battery recycling system.

**Table 1 ijerph-20-04346-t001:** Relevant regulations on EoL power battery recycling in the EU.

	Attributes of Different Policies	
Date	Name	Key Points of Policy
Mar 1991	Council Directive 91/157/EEC	Regulation of the mercury content in batteries.
Sept 2000	End of Life Vehicle Directive 2000/53/EC	Reduce waste from vehicle scrapping and increase the recycling rate of EoL vehicles. Recycling costs are mainly paid by the manufacturers.
Sept 2006	Battery Directive 2006/66/EC	Battery manufacturers establish EoL battery recycling systems, increase the recycling of EoL battery products, and improve the treatment and recycling rate of EoL batteries.
Dec 2013	Battery Directive 2013/56/EU	Clarify the requirements for hazardous substance content, recycling labels, and hazardous substance labels.
Dec 2020	New Battery Regulation 2020/0353 (COD)	Waste Battery Directive 2006/66/EC, focusing on the whole life cycle treatment of EoL batteries.

**Table 2 ijerph-20-04346-t002:** Federal regulations on EoL power battery recycling in the U.S.

	Attributes of Different Policies		
Date	Name	Ministry	Key Points of Policy
Oct 1976	Resource Conservation and Recovery Act	EPA	Amends the Solid Waste Disposal Act enacted in 1965 and proposes a basic framework for hazardous waste management.
May 1996	Mercury-Containing and Rechargeable Battery Management Act	EPA	Supports the collection and recycling of rechargeable batteries containing heavy metals such as lead and cadmium.
Jun 2021	National Lithium Development Blueprint 2021–-2030	DOE	Analyzes the current development status and prospects of the U.S. lithium industry and proposes the construction goals of the lithium battery supply chain.
Feb 2022	Battery Materials Processing and Battery Manufacturing Funding Opportunity Announcement (DE-FOA-0002677)	DOE	Provides approximately USD 2.8 billion to fund R&D in EV battery material processing and battery component manufacturing and recycling.
Feb 2022	Electric Drive Vehicle Battery Recycling and Second-Life Applications (DOE-FOA-0002679)	DOE	Provides approximately USD 60 million to fund research and development, testing of EV battery recycling, stepping-up, reusing to scale, and profitability.

EPA: U.S. Environmental Protection Agency; DOE: U.S. Department of Energy.

**Table 3 ijerph-20-04346-t003:** Laws on EoL power battery recycling in Japan.

	Attributes of Different Policies	
Date	Name	Key Points of Policy
2000	The Basic Law for Establishing the Recycling-based Society	Establish the basic principles of waste and recycling policy.
2002	The Law for the Promotion of Effective Resource Utilization	Specify the product producer’s responsibility and recycling system construction and other requirements and emphasize the recycling of resources.
2003	The Law for the Recycling of End-of-life Vehicles	Clarify the cost of end-of-life vehicle treatment specifications and resource recovery and reuse.

**Table 4 ijerph-20-04346-t004:** Laws and regulations on EoL power battery recycling in China.

	Attributes of Different Policies		
Date	Name	Ministry	Key Points of Policy
Feb 2006	Technical policy on recycling and utilization of automobile products	NDRC	The first policy of recycling scrap auto parts based on the EPR system puts forward three-phased goals for vehicle product recycling and utilization.
Aug 2008	Circular Economy Promotion Law	CPG	Dismantling or reusing waste lead–acid batteries should comply with the law.
Jan 2016	Technical Policy on Electric Vehicle Power Battery Recycling (2015 Version)	NDRC	Guides the orderly recycling and utilization of EV power batteries, promotes resource regeneration, and puts EPR into effect.
Feb 2018	Interim Measures for the Management of New Energy Vehicle Power Battery Recycling	MIIT	NEV manufacturers must implement the EPR system and assume primary responsibility for recycling batteries. They must adhere to the battery’s life cycle concept and the principle of the organic unity of environmental, social, and economic benefits. They must Ensure that the battery can be effectively recycled and treated environmentally responsibly.
Jul 2018	Interim Regulations on Traceability Management of New Energy Vehicle Power Battery Recycling	MIIT	Establish a comprehensive management platform for power battery recycling and carrying out information collection and responsibility monitoring for all aspects of recycling.
Nov 2019	New Energy Vehicle Power Battery Recycling Service Network Construction and Operation Guidelines	MIIT	Vehicles and echelon utilization companies build shared recycling service stations and enhance tracking of EoL power batteries.
Jan 2020	New Energy Vehicle EoL Power Battery Comprehensive Utilization Industry Specification Conditions (2019 Version), New Energy Vehicle EoL Power Battery Comprehensive Utilization Industry Specification Announcement Management Interim Measures (2019 Version)	MIIT	Defines the meaning of comprehensive utilization and divides it into echelon utilization and recycling. Standardizes the technical, production, and environmental requirements for enterprises and strengthens the supervision and management of enterprises. Proposes work, construction, safety, and environmental protection requirements.
Aug 2021	Management Measures for the Echelon Utilization of New Energy Vehicle Power Batteries	MIIT	Strengthens the management and quality of the echelon utilization of NEV power batteries and standardizes the recycling and environmental protection disposal of secondary utilization enterprises. The companies add agreement cooperation and information sharing and use existing recycling methods to recycle efficiently.

Ministry of Industry and Information Technology (MIIT); The Central People’s Government (CPG); National Development and Reform Commission (NDRC).

## Data Availability

Not applicable.

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
