# Peer review of "Literature Review on Power Battery Echelon Reuse and Recycling from a Circular Economy Perspective"

_ijerph, 2023, doi:10.3390/ijerph20054346_

Round 1

Reviewer 1 Report

I would like to thank you for inviting me to review this manuscript. Overall, I thought it was a strong paper. I felt the research topic was interesting and the article appeared well-written and well-referenced. However, I am recommending that the paper is accepted for publication with minor revisions.

We need some information about the types of studies selected by the author and the methods around the article selection.

I wonder if this study was not associated with any limitations. Therefore, my suggestion is to mention the limitations of the study in the discussion section. When addressing limitations, your future directions should also be tailored to the limitations identified. For each limitation, there should be a future direction that addresses it.

Author Response

Thanks for your guidance for our manuscript entitled “Literature Review on Power Battery Echelon Reuse and Recycling from a Circular Economy Perspective” (ID: ijerph-2174063). Your comments on this manuscript provide an exact direction for further improvement.

With your meticulous guidance, we were able to conduct a more in-depth study and bring about a qualitative improvement in the paper. All authors would like to express their sincere appreciation to you for your help all the way.

The one-to-one response to your comments are uploaded along with the manuscript.

Reviewer 2 Report

The second point in line 91 is repeated with the third point in line 93.

The four subheadings on page 14 should correspond to the word order of 465 lines

Author Response

(The authors gave the same response as above.)

Reviewer 3 Report

This paper first analyzes representative countries’ power battery recycling policies and finds the reasons for the low recycling rate in some countries. And this paper analyzes the existing policy problems and existing technical challenges. Based on the actual situation and future development trends, we propose development suggestions from the government, enterprises and consumers to achieve the maximum reused of end-of-life (EoL) power batteries. In my opinion, it is acceptable after revision. The comments and suggestion are listed below.

1、  In fig 2, There is something wrong with the ordinate unit of the graph.

2、  In fig.8, The font of the picture is so small that it affects reading.

3、  In order to reflect the authenticity of the data.The data in this paper should provide references. eg fig 1and fig 2 et al.

4、  The format of references is not uniform. eg 44 et al.

Author Response

(The authors gave the same response as above.)

Reviewer 4 Report

The manuscript needs substantial changes to be accepted for publication

1-The title can be restructured to provide more clarity to the readers

2-In the Introduction kindly provide the need of sustainable mobility, refer

https://www.mdpi.com/2071-1050/13/22/12918
3-The statement "It is expected that by 2030, the proportion of new energy3
and clean energy-powered transport vehicles will be around 40%. " needs reference
4-More recent references can be amended. Figures needs to be high resolution and described in depth

Comprehensive proof reading is mandate

Author Response

(The authors gave the same response as above.)

Reviewer 5 Report

The paper presenting a comprehensive work on power Battery Echelon Reuse and Recycling from a Circular Economy Perspective. Overall, the writing is comprehensive and general, but I felt it is lacking substance and in-depth statistical analysis.

1.     In the conclusion, it is more appropriate to write as: This paper summarizes the four existing recycling "models": battery manufacturers ...

2.     I would suggest authors to come out with more statistical analysis & review.

3.     A more details on the costing for recycling.

4.     More elaboration and details analysis based on current recycling situation. For example, what is the current recycling status (cost, program, quantity recycle cells, profit etc.) for BYD, Trumpchi, FAW-Volkswagen and Geely.

Author Response

(The authors gave the same response as above.)

Reviewer 6 Report

“Literature Review on Power Battery Echelon Reuse and Recycling from a Circular Economy Perspective”

This work is well written and discussed. However, there are some drawbacks that need to get addressed before getting it published:

·       It is not clear how this study would bridge any gaps in the relevant literature. It should be mentioned clearly in the introduction section.

·        To improve the quality of the introduction section, the following article is suggested:

(1) “A multigeneration cascade system using ground-source energy with cold recovery: 3E analyses and multi-objective optimization” https://www.sciencedirect.com/science/article/abs/pii/S036054422101433X

Author Response

(The authors gave the same response as above.)
